# Mechanical Forces as Determinants of Disseminated Metastatic Cell Fate

**DOI:** 10.3390/cells9010250

**Published:** 2020-01-19

**Authors:** Marco Montagner, Sirio Dupont

**Affiliations:** Department of Molecular Medicine, University of Padua, via Bassi 58/B, zip 35121 Padua, Italy

**Keywords:** ECM, MRTF/SRF, YAP/TAZ, dormancy, forces, integrins, mechanotransduction, metastasis

## Abstract

Disseminated metastatic cancer cells represent one of the most relevant causes of disease relapse and associated death for cancer patients, and a therapeutic target of the highest priority. Still, our understanding of how disseminated cancer cells survive in the foreign metastatic environment, and eventually cause metastatic outgrowth, remains rather limited. In this review we focus on the cell microenvironment as a key regulator of cell behavior at the metastatic site, and especially on the mechanical properties of the extracellular matrix and associated integrin signaling. We discuss available evidence pointing to a pervasive role of extracellular matrix (ECM) mechanical properties in regulating cancer cell proliferation and survival after dissemination, and propose that this might represent an important bottleneck for cells invading and establishing into a novel tissue. We point to the known molecular players, how these might contribute to modulate the mechanical properties of the metastatic environment, and the response of cells to these cues. Finally, we propose that emerging knowledge on the physical interaction of disseminated metastatic cells and on the downstream mechanotransduction pathways, including YAP/TAZ (Yes-associated protein-1 and WW-domain transcription activator 1) and MRTFs (Myocardin-related transcription factors), may help to identify novel approaches for therapy.

## 1. Introduction

Cancer cells, like any other cell in our body, live in a complex microenvironment made of other cells, the extracellular matrix, and soluble molecules that diffuse in the interstitial fluids. Although cancer initiation is unmistakably driven by genetic lesions hitting oncogenes and tumor suppressors, there is increasing evidence that the tumor microenvironment plays key “epigenetic” role in dictating whether or not a cell bearing an oncogenic mutation will develop into a cancer [1]. Moreover, the cell microenvironment not only influences primary tumor growth, but also affects the ability of cancer cells to resist chemotherapy, to migrate away from the primary site, and to establish secondary metastatic foci. As a result, it is an accepted general notion that studying the cancer microenvironment might provide insights into the mechanisms driving cancer progression, and the basis for developing new therapeutic approaches. The relevant question then becomes which feature of the microenvironment is relevant, and to which part of cancer biology. In this review, we chose to focus on cellular mechanosensors and on the mechanical properties of the ECM, which are important features that can regulate cell behavior, but whose role is often neglected, especially in the metastatic context.

## 2. ECM Mechanical Forces in Cancer

Mechanical forces are ubiquitous in tissues, and profoundly affect cell behavior. Although we may think of forces as limited to organ systems inherently participating in force bearing or production (the circulatory system, the musculoskeletal system, and the respiratory system), forces are a main ingredient of many biological processes such as cell division, the formation of cell protrusions, cell migration, and tissue morphogenesis [2,3]. Even more surprisingly, it is now evident that forces can also influence more general processes including cell proliferation, differentiation and death by regulating intracellular signaling pathways and gene transcription, similar to a cytokine or extracellular growth-factor treatment [4,5,6,7].

It is now widely recognized how cancer cells experience a “force journey” during the progression of the primary tumor, invasion in neighboring tissues, and dissemination to distant metastatic sites (Figure 1) [8,9]. During this journey, cancer cells face multiple microenvironments imposing different mechanical constraints. In situ cancer cell growth increases intratumoral pressure (Figure 1b). Many cancer types, alone or in collaboration with stromal cells, remodel the ECM to decrease its tumor-suppressive features (for example, by degrading the soft basal membrane that prevents epithelial cell dissemination) and to favor its tumor-promoting ones (for example, by increasing stiff cross-linked collagen content and by orienting collagen fibers around the tumor to favor outward cancer cell migration) (Figure 1c). Seminal works indicate that the tumor ECM is not only important to promote cell invasion, but plays a broader role to enable the expression of an oncogene’s transforming potential [10,11,12,13]. Indeed, it was shown that the transformed phenotype displayed by cancer cells in standard 2D tissue culture conditions is reverted to a non-tumorigenic phenotype when the same cells are embedded in the same ECM, but in a 3D setting. One of the main differences between the two conditions is the stiffness or elasticity of the ECM, dictated by plastics or glass in 2D (very stiff), while in 3D this depends on the composition and arrangement of the ECM molecules themselves (usually, much softer).

Then, once a cancer cell reaches blood or lymph vessels it faces the physical barrier posed by endothelia (Figure 1d) to enter in a completely new environment characterized by lack of adhesion to the ECM (Figure 1e). While in the blood, cancer cells will experience shearing forces from blood flow, which will influence their ability to adhere to blood vessel walls (Figure 1e) and extravasate (Figure 1f) [14]. Finally, past this last physical endothelial barrier, cancer cells will enter an alien microenvironment characterized by mechanical forces that can be very different from those present in the primary tumor site (Figure 1g), and that in the most common cases of solid tumor metastasis (i.e., to bone marrow, lung, liver and brain) can be softer compared to the primary tumor ECM [8,9]. This may impose a strong selective pressure for the settling of cells at the metastatic site and may regulate disseminated cancer cell behavior including dormancy and resistance to chemotherapy.

## 3. Cells Measure the Physical Properties of the ECM

The process by which cells sense an external mechanical stimulus and convert it into an intracellular biochemical response is generally defined as mechanotransduction (Figure 2) [5,15]. Mechanotransduction can be broken down into a sequence of events: (1) the transmission of force to specialized cell structures (mechanotransmission); (2) the conversion of force into a signal of biochemical nature (mechanosensing); (3) and the subsequent response of the cell to that signal (mechanosignalling or mechanoresponse), which may or may not be shared with other types of signals. This last part deals both with the signaling cascade activated as a result of forces, and with the more downstream phenotypic responses (alteration of gene transcription, alteration of proliferation or survival, alteration of metabolic parameters etc.).

How then do cells measure ECM elasticity? The structures that enable force transmission in cells are the integrin receptors, which physically connect the ECM with the cytoplasm, the focal adhesions (FAs), and the actin cytoskeleton anchored to FAs [5,16]. These structures not only transmit forces, but actively respond to force loading, such that integrin clustering, focal adhesion maturation and F-actin stress fiber formation are proportional to the resisting visco-elastic forces exerted by the ECM. The current tenet to explain the mechanosensitive ability of cells poses that cells contacting the ECM initially form small transient contacts by local clustering of integrins (Figure 2a), which start the formation, on the cytoplasmic side, of immature focal adhesions (or focal points) that connect to F-actin (Figure 2b). Then, cells start actively probing the elasticity of the ECM by developing internal tension or contractility, thanks to the activity of the ubiquitous non-muscle myosin isoforms (NMII) on F-actin bundles (Figure 2c): if extracellular resistance is high enough, tension across the system will stretch some proteins, such as Talins or p130cas, enabling the growth of focal contacts into mature focal adhesions, accompanied by enlarged F-actin bundles and enhanced contractility (Figure 2c–f); if extracellular resistance is low, tension will disengage integrins from the ECM and focal adhesion maturation is prevented [5,16]. This dynamic process allows a quantitative response of focal adhesions themselves to external applied forces, and is tuned to respond across a physiologically-relevant tissue stiffness range (i.e., of Young’s modulus between 10^2^ and 10^5^ Pascals).

The above model predicts that inhibition of any of the players involved in the formation of focal adhesions and in enabling cell contractility (including NMII, ROCK and MLCK kinases, RHO GTPases and formins; see Figure 2c) will induce cells to believe they are experiencing a soft ECM even if the ECM is actually very stiff, a hypothesis that has been extensively validated experimentally [5,15]. Another corollary of this model is that cell geometry is intertwined with cells’ mechanosensing ability: when cells attach to a stiff ECM, they will not only mature their FAs, but they also spread to gain space and develop longer and stronger F-actin bundles, such that projected cell area is often proportional to ECM elasticity. On the other hand, it is sufficient to limit cell spreading, even in the presence of a stiff ECM, to prevent cell contractility, maturation of FAs, and the corresponding biological responses [5,17].

One important missing point in this model is what happens when cells are in a 3D ECM. Multiple evidence suggests that FA and F-actin dynamics are different in 2D and 3D conditions, such that dimensionality may change the way cells perceive the ECM [9,18]. This is particularly true in the case of cell migration, which is almost entirely dependent on FAs in 2D, but can happen in absence of any mature FA, or even without any specific ECM adhesion, in 3D [19]. However, it must also be considered that many cell types continuously change shape while they migrate, even on a 2D substrate, but this does not appear to affect their ability to sense the overall stiffness of the ECM–which would suggest that cells are able, to some extent, to uncouple transient cell shape changes occurring during migration from ECM mechanotransduction, maybe due to the different time-scales at which these processes occur. Moreover, experimental data indicate that the general rules that hold true in 2D also apply to 3D growth, such that ECM stiffness regulates similar phenotypes despite dimensionality. In line, some players that regulate ECM mechanosensing (p190 RHO GTPase, CAPZ, ROCK inhibitors) have a coherent biological activity in 2D and in vivo, suggesting that the basic model holds generally true also in tissues [20,21,22]. In other instances, a stiff 3D ECM induces phenotypes that are similar to a soft 2D [23,24]; these are only superficially counterintuitive, because they can be explained by a “confinement” effect due to the inability of cells to degrade the 3D ECM and dig enough space to spread and develop contractility, similar to what happens on a small and stiff 2D ECM. Finally, it must be considered that different types of ECM (such as fibronectin, collagens or laminins), the 3D organization of these proteins, and the different types of integrin receptors themselves may modulate the perception of mechanical forces by cells, as well as having different ability to activate other pathways, in parallel to mechanotransduction.

Another general consideration is that many cells in tissues such as epithelia are not found isolated but are tightly connected to neighboring cells via cell-cell junctions. This is true even for many migrating or invading cells, which undergo collective migration. Cell–cell adhesion structures also connect, directly or indirectly, to contractile actin bundles, can act as force transducers, and actually respond to extracellular force application by growing [25]. At least in part, this process depends on force-mediated unfolding of alpha-catenin, at the very heart of adherens junctions, and force-dependent recruitment of actomyosin fibers to cell–cell junctions [26]. This mechanism is thought to increase the resistance of epithelia subjected to stretching forces, preventing tissue ruptures. However, even if the basic mechanisms appear extremely similar to those involved in mechanotransduction at FAs, the biological effects appear different if not opposite: in several epithelial tissues, reduced integrin adhesions induce reduced proliferation, while reduced cell-cell adhesion has the opposite effect. Moreover, data suggest that even cells organised into an epithelium do sense differential ECM stiffness [4]. Thus, forces at cell-ECM adhesions and forces at cell–cell junctions are differentially sensed and evoke different responses.

## 4. Downstream Pathways Modulated by Forces

Downstream or integral to the ECM-integrin-actin mechanosensitive system, mechanisms must exist that translate forces into biochemical signals, in turn instructing cell behavior. Among the known or proposed mechanisms, the simpler entails force-dependent unfolding of proteins connected to FA or to the cytoskeleton [27]. Talins and p130cas are a point in case (Figure 3a), as they can recruit and activate other proteins in response to force application [28,29]. So far, these interactions have been shown to play a role at the heart of the FA maturation process itself [5,16], and it remains unclear whether these also pass the signal to more downstream signaling events, or if additional proteins play a specific signaling role. Among the FA-regulated proteins, focal adhesion kinase has been an interesting candidate to mediate the signaling effects of forces (Figure 3a), but its effects on mechano-regulated phenotypes indicates FAK is either an ancillary mechanism [30], or a protein actually limiting stiffness-induced phenotypes [31]. Other interesting candidates are the Src kinases and the RAP GTPases (Figure 3a), whose activity can be modulated by extracellular forces [32,33].

A second type of responses occur by direct transmission of forces through the actin cytoskeleton to other cell structures, including the plasma membrane or organelles. Deformation or stretching of the plasma membrane activates mechanosensitive ion channels such as the Piezo1/2 cation channels (Figure 3b) [34]. These effects are not limited to neural or specialized sensory cell types, but appear widespread in many cell types and tissues, including cancer cells. In the context of ECM stiffness, Piezo1/2 channels are activated at nascent focal adhesions to reinforce their maturation, and would thus represent an additional mechanism to reinforce force-mediated FA maturation [35,36]. Another mechanosensitive process at the plasma membrane is the formation of caveolin-dependent endocytic vesicles [37], that provide cells a membrane reservoir to quickly accommodate sudden stresses (Figure 3b). This mechanism has not been demonstrated in the context of ECM elasticity. However, caveolae play a role in sustaining actomyosin contractility, ECM remodeling and the responses to ECM stiffness [38,39].

Direct transmission of forces to intracellular structures (Figure 3c) has been initially reported for the nucleus [6]. The nucleus is physically connected to cytoplasmic F-actin through LINC complexes, and nuclear shape is modulated by differential adhesion and spreading onto a soft vs. stiff ECM. Moreover, LINC complexes cross the nuclear envelope and connect with the lamin nucleoskeleton. This allows extracellular forces to control LaminA phosphorylation and turnover, which influences the expression of cytoskeletal genes and cell differentiation [40,41]. Structurally, this mechanism increases nuclear envelope rupture in cells on a stiff ECM, which in absence of efficient DNA repair mechanisms causes DNA damage [42,43]. Finally, the link with LaminA may also mediate direct force transmission to chromatin, and thus influence chromatin dynamics and epigenetic regulation of gene expression [44]. Other cytoplasmic organelles that can respond to F-actin contractility and extracellular forces are the endoplasmic reticulum (ER) and the Golgi apparatus [45,46,47]. In the latter instance, this links ECM stiffness to lipid metabolism [47,48]: decreased actomyosin tension experienced on a soft ECM inhibits the activity of the Lipin-1 phosphatidate phosphatase, resulting in decreased activity of the ARF1 GTPase. In turn, decreased ARF1 activity alters the distribution of the SREBP1/2 precursor molecules between the ER and the Golgi apparatus, favoring their proteolytic processing at the Golgi and the translocation of their cytoplasmic domains in the nucleus, ultimately increasing transcription of enzymes involved in lipid and cholesterol synthesis. Surprisingly, in this case SREBP lipogenic activity is skewed specifically towards neutral lipid synthesis [47], at difference with oncogenic stimulation of SREBP favoring the production of new membranes [49]. It thus remains to be determined what is the functional significance of this metabolic phenotype, in physiology and in cancer.

A third type of response to mechanical forces is mediated by transcription factors, accounting for activation of specific gene-programs downstream of ECM or tissue mechanics (Figure 3d). One of such factors is the MRTF-SRF complex. MRTFs are paralog transcriptional coactivators that shuttle between the cytoplasm, where they are inactive, to the nucleus, where they bind to the SRF DNA-binding factor and regulate transcription [50]. MRTFs bind to monomeric G-actin, which precludes their nuclear accumulation. MRTFs are thus activated upon F-actin polymerization, for example upon activation of RHO mediated by serum stimulation [51]. Given that RHO activity and F-actin polymerization are also generally favored when cells are cultured on a stiff ECM, MRTFs can be also regulated by mechanical forces [52,53,54]. Recent studies on MRTF-driven genetic programs indicate that a main function of these factors is to reinforce the contractile actomyosin machinery [55], which likely empowers and stabilizes cell responses to ECM stiffness.

The other general mechano-responsive transcription factor identified so far is the YAP/TAZ-TEAD complex. Similar to MRTFs, YAP/TAZ are paralog transcriptional coactivators that shuttle between the cytoplasm, where they are inactive, to the nucleus, where they bind to the TEAD family of DNA-binding factors and regulate transcription [56]. At difference with MAL/MRTFs, however, YAP/TAZ do not directly bind actin. Several mechanisms have been proposed to explain the regulation of YAP/TAZ by ECM stiffness, including: (1) nuclear F-actin competing away ARID1A from YAP/TAZ binding, thus setting nuclear YAP/TAZ free to bind TEAD factors [57]; (2) cytoplasmic F-actin inducing deformation of the nuclear envelope and thus opening of the nuclear pore complexes, in turn empowering nuclear translocation of YAP/TAZ [58,59]; (3) inhibition of the LATS1/2 Hippo kinases downstream of RAP2 GTPase activation at focal adhesions [32]. It is likely that these mechanisms cooperate to regulate YAP/TAZ acting at different and sequential steps, i.e., nuclear accumulation and transcriptional activity, thus ensuring a robust regulation. YAP/TAZ mediate a plethora of transcriptional responses to the mechanical properties of the ECM and to cell geometry, including: mesenchymal stem cell, keratinocyte and pancreatic progenitor differentiation; endothelial proliferation vs. apoptosis; fibroblast activation and senescence; organoid growth in defined matrices; cancer cell proliferation [4]. While it is now generally recognized that proliferation (including factors promoting G1/S and G2/M transition, DNA repair mechanisms and enzymes for dNTP synthesis) represents one of the main genetic programs activated by YAP/TAZ in multiple cell types [60,61,62,63], how YAP/TAZ regulate cell differentiation remains less characterized, although a main emerging theme is the regulation of Notch signaling [64]. Another phenotype induced by YAP/TAZ and relevant for cancer biology is epithelial-to-mesenchymal transition (EMT) [65,66], which may be further reinforced by ECM stiffness owing to the direct regulation of Twist1 nuclear localization [67]. Importantly, YAP/TAZ can be regulated not only by ECM stiffness but also by other types of forces, such as cell stretching or extracellular fluid shear stresses [4,68,69]. The link between mechanical forces and YAP/TAZ is particularly interesting for cancer biology, because of the powerful effects of the ECM and because genetic data in the mouse indicate that YAP/TAZ are dispensable for the homeostatic self-renewal and proliferation of multiple tissues, while they are strongly and generally required for cancer cell proliferation and survival [70]. YAP/TAZ thus appear as ideal molecular targets to disable cancer cell survival while sparing normal tissues.

## 5. Mechanical Forces as Regulators of Disseminated Cancer Cells’ Fate

A formal proof that survival, quiescence or reawakening of disseminated metastatic cancer cells are regulated by mechanical forces is still missing. Likely, this is due to technical limitations in our ability to address the functional relevance of forces in vivo. Yet, ECM composition as well as mechanosensing machineries and transducers deeply influence the outcome of the metastatic process (Table 1). For example, it has been recently proposed that metastatic tropism, the organ-specific pattern of metastasis of each primary tumor, is in part determined by the stiffness of the target organ. So far, the preference of a certain tumor type for a specific secondary organ has been explained with two, non-mutually exclusive, hypotheses: Paget’s “seed and soil” hypothesis, and the “mechanical” hypothesis. The former posits that disseminated cells will preferentially seed organs with a favorable microenvironment (the soil), emphasizing the importance of ECM composition, soluble signals, supporting stromal cells or nutrients availability. The latter explains metastatic tropism based on anatomy of blood vessels and dissemination routes [9]. Determinants of metastatic tropism are both microenvironmental-driven and genetically encoded, as demonstrated by the works of the Massague lab with single cell populations (SCPs), derived from the same breast cancer cell line, with different target specificity [71]. Despite the differential tropism of these cells has been explained with a variety of mechanisms [72,73,74,75], recently it has been proposed that matrix rigidity of the target organ may contribute as well to preferential seeding of cancer cells [76]. Kostic et al. demonstrated that MDA-MB-231-derived SCPs with different tissue tropism, showed increased growth and migration in ECMs with the same rigidity of the organs where metastases are observed. The instructive role of three-dimensional architecture has been investigated by the Green lab in their pioneering work on metastatic dormancy, where they showed that cancer cells embedded in a 3D environment in vitro adopt a proliferative behavior similar to their dormant phenotype at the metastatic site in vivo [77,78]. D2.A1 and D2.0R cells are two isogenic cell lines with comparable growth rate in vitro, but striking different behavior in vivo, with A1 cells forming overt metastasis in the lungs and 0R persisting as dormant colonies, upon tail vein injection [79]. By leveraging the protocol developed by the Brugge lab for growing mammary epithelial acini [80], Green and colleagues showed that upon cultivation of cells within growth factor–reduced three-dimensional basement membrane extract at low concentration, D2.0R cells persisted as small quiescent clusters, while D2.A1 cells rapidly formed invasive spheroids, recapitulating their behavior in vivo. This in vitro system has been further exploited by the Weinberg group to describe the importance of filopodium-like protrusions in the outgrowth of aggressive cells vs. non-metastatic cells [81,82].

## 6. Extracellular Matrix Proteins and Metastatic Outgrowth

Several fibrous and non-fibrous ECM proteins play a role in each and every step of metastatic colonization (Table 1). Interestingly, ECM proteins such as periostin, tenascin C, lumican and fibronectin can be secreted by disseminated cells or by resident reactive stromal cells, following tumor-stroma crosstalk [83,84,85], suggesting that irrespectively of the source, these proteins may increase the fitness of metastatic cells. Fibrous ECM proteins include collagens, laminins and fibronectin. Altering collagen type I organization and deposition deeply affects the tensional homeostasis of untransformed cells and predispose to cancer development. Thick and aligned collagen fibers have been observed around primary tumors where they contribute to stroma stiffening and dissemination of cancer cells [86,87,88], it is not surprising then that fibrotic conditions (such as cystic fibrosis and cirrhosis) significantly increase the risk of malignancy [87]. Importantly, increased collagen type I in the lung parenchyma leads to dormant-to-proliferative switch of quiescent metastatic breast cancer cells. Response to collagen I is mediated by integrinβ1 activation of Src, FAK and ERK with phosphorylation of MLC (pMLC) and can be mimicked in vitro by adding collagen I to the ECM [78]. Although not experimentally tested, mechanical forces are likely to play an important role in this context. Beyond the amount of collagen, crosslinking of collagens by lysyl oxidases (LOX) increases their tensile properties and stiffness, which has been linked to augmented local invasion of premalignant epithelium [10] as well as pre-conditioning of the metastatic niche and increased metastasis [8,89,90]. Laminin is another ECM fibrous protein recently linked to exit from metastatic dormancy. Cleavage of laminin by inflammation-triggered neutrophils extracellular traps (NETs) in the lung ignites proliferation of otherwise quiescent breast cancer cells through integrin activation and FAK/ERK/MLCK/YAP signaling [91]. Type I and IV collagens and fibronectin have been shown to support the survival of breast cancer cells in an in vitro model of dormancy bone marrow [12,92]. Fibrillar fibronectin deposited by cell populations under serum-deprivation mediates the survival under quiescence of those cells via αvβ3 and α5β1 integrin adhesion, ROCK-generated tension, and TGFβ2 stimulation [93]. Degradation of fibronectin by MMP-2 is required for cancer cells outgrowth in vitro [93], which parallels the findings that mesenchymal stem cells and intestinal crypts embedded in 3D matrices require MMP-mediated remodeling to avoid spatial confinement, and thus to avoid mechanical inhibition of YAP activity [23,24]. Moreover, fibronectin was also shown to regulate P-ERK/P-p38 ratio, a well-established driver of proliferative/quiescent switch, in primary tumor dormancy [94,95], and a well-established response of cells to the mechanical properties of the ECM [30,96].

Another key determinant of the response of cells and tissues to mechanical stress is the glycocalyx, i.e., the shell of polysaccharides and glycoproteins that decorate the exterior of the cell surface, which provides hydration to extracellular space and, thus, resistance to compressive forces. Far from being an inert component of cell surface, the composition of the glycocalyx is dynamically regulated, and this is important for several aspects of tissue homeostasis. In cancer, bulky glycoproteins such as mucin-1 are enriched in advanced tumors and sustain survival of CTCs [97,98]. Interestingly, this activity does not depend on a signaling function of glycoproteins, but rather on a physical/sterical effect that promotes local integrin clustering and downstream activation of the PI3K/Akt pathway, such that it can be mimicked by synthetic bulky glycopolymers or by expressing a signaling-defective but glycosylation-proficient Mucin-1 isoform [97,98].

Non-fibrous proteins are another essential component of ECM because they modulate the interaction of membrane receptors with ligands or other ECM proteins, examples of these proteins are periostin (POSTN), tenascin C (TNC) and thrombospondin 1 (TSP-1). Tenascin C is a glycoprotein expressed in connective tissues and neural crest cells, but it’s found upregulated in several cancers with a negative link with prognosis [84,93,99]. The association with increased lung metastasis in patients led Oskarsson and colleagues to investigate its role in the metastatic process. They discovered that TNC supports survival and metastasis-initiating abilities of disseminated breast cancer cells by enhancing stem cells signaling pathways such as Notch and Wnt [75]. Another glycoprotein, TSP-1, has been linked to control of quiescence of disseminated breast cancer cells [83,91], although its contribution to cancer development seems highly context-dependent [100]. In an in vitro model of dormancy, TSP-1 is secreted by the mature endothelium within lung and bone perivascular niches and maintains the quiescence of breast cancer cells. When neovascular sprouting is induced, loss of TSP-1 and secretion of new ECM proteins (POSTN, TNC, TGFβ1) trigger reawakening of cancer cells [83]. POSTN regulates ECM structure and organization by acting on collagen I cross-linking via BMP-1/LOX axis and it is upregulated in a number of cancers [101]. Additionally, POSTN increases the fitness of lung-disseminated breast cancer stem cells by recruiting Wnt ligands [85].

## 7. Mechanosensing Receptors and Metastatic Fitness

In addition to ECM proteins, several members of mechanosensing machineries (such as integrins, discoidin domain receptors (DDRs), syndecans, growth factor receptors and stretch-activated ion channel) have been implicated in metastatic dormancy or proliferation (Table 1), indicating that cancer cells can mold the perception of the 3D architecture rather than the physical properties of the environment itself. The importance of integrins in any step of the metastatic cascade has been extensively described in excellent reviews; in this piece we will focus on integrins as an emerging ubiquitous theme in the regulation of metastatic dormancy. Initial evidences of integrin involvement into regulation of proliferation after extravasation were provided by Green and colleagues in their early works with three-dimensional matrices. By comparing cell lines with different metastatic potential in vivo, they showed that quiescence-to-proliferative switch of aggressive cells correlates with integrin b1 activation, stress fibers formation and phosphorylation of MLC [77]. Importantly, they reported that indolent disseminated cancer cells are not doomed to growth arrest as they can promptly re-enter active proliferation if integrin-β1 is engaged by fibrotic collagen I and activates FAK/Src/ERK pathway in vitro and in vivo [78]. The same conclusions have been further reinforced and deepened by Weinberg and colleagues using the same in vitro model and several D2.A1 mutants defective for components of adhesion and mechanotransduction complexes. They observed that metastasis-proficient cells develop a higher number of dynamic filopodia-like protrusions (FLPs) compared to dormant cells. Combined action of Rif/mDia2 and ILK/b-parvin/cofilin are required for formation of FLPs and metastatic outgrowth [81,82]. Similarly, the inhibition of the Fascin actin-bundling protein, enriched at and required for filopodia formation, can affect metastatic cell survival [102,103,104]. Interestingly, cofilin inactivation promotes YAP activity [105], which may suggest a role for the mechanoregulated factor YAP in these phenotypes. In the bone perivascular niche, α_5_β_3_ and α_4_β_1_ integrins are stimulated by von Willebrand Factor (VWF) and VCAM-1, respectively, from the endothelial cells and actively sustain chemoresistance of bone-disseminated breast cancer cells [106]. Again, blocking these integrins in combination with doxorubicin and cyclophosphamide proved to be a valuable strategy to circumvent chemoresistance in vivo [106]. Inflammation is a clinically relevant trigger of relapse of metastatic breast cancer. Smoke-induced inflammation in the lungs recruits neutrophils that remodel the ECM by secretion of NETs and proteolytic cleavage of laminin. Once cleaved, laminin exposes an epitope that stimulate α_3_β_1_ integrin signaling, FAK/ERK/MLCK axis activation and growth of otherwise dormant breast cancer metastasis [91]. Integrinβ1 and ILK can be also activated by L1CAM expressed by disseminated cancer cells, and are responsible for the pericyte-like spreading of cells upon induction of YAP and MRTF protumorigenic pathways [107]. Integrins in stromal cells also have an important role in driving outgrowth of indolent metastatic lesions. Bone-latent breast cancer cells expressing VCAM-1 recruit α_4_β_1_ integrin-positive monocytic osteoclast progenitors that elevate local osteoclast activity favoring metastasis formation [73]. Integrins can also couple with growth factor receptors, such as EGFR, and activate signaling pathways such as PI3K/Akt and increase fitness of cancer cells [12].

DDR proteins, receptors for collagens, can cooperate with other membrane receptors to regulate metastasis outgrowth. Starting from a genetic screening in vivo, the Giancotti group showed that DDR1 couples with TM4SF1, an atypical tetraspanin receptor, and activates PKC-JAK-STAT3 leading to metastatic outgrowth of indolent cancer cells in lung and bone [108]. Transmembrane heparan sulfate proteoglycans, such as syndecans, expressed in disseminated cells has been recently shown to repress proliferation at the metastatic site via recruitment of the PARD3/PARD6/atypical-PKC protein complex and release of Par-1 kinases into the cytosol. Cytosolic Par-1 inhibits KSR scaffolding proteins and thus Ras/ERK signaling, important determinants of exit from dormancy. On the contrary, mice lacking Syndecan-1 in the lung stroma showed temperature-dependent decreased metastasis from intravenously injected breast cancer cells [109].

## 8. Mechanosensing Pathways and Dormancy at the Metastatic Site

Extracellular cues are transduced within cells by signaling cascades that eventually converge at the level of gene transcription. Mechanical forces do not translate into intracellular signals by dedicated signaling pathways, instead, they use the same signaling routes exploited by biochemical signals. In order to do that, cells evolved several strategies to transform a physical stimulus into a biochemical event: regulation of receptors availability, clustering of molecular scaffolds, conformational-controlled enzymatic activity, or even ligand processing and cytoskeleton-regulated molecular transducers. In this section we will summarize how the main signaling pathways involved in the dormant phenotype (Table 1) can be modulated by physical cues such as stiffness of the substrate, as well as compressive or tensional stresses.

TGFβ is a pleiotropic family of cytokines, it takes part essentially in every cellular and physiological process from early stages of embryonic development, maintenance of tissue homeostasis to cell death and its dysregulation is at the origin of several diseases, including inflammation, fibrosis, degenerative conditions and malignancies. TGFβ has a dual role in cancer progression: it is known to be a strong inducer of growth arrest during homeostasis or at the early stages of transformation, but it can exacerbate the malignant phenotype in the advanced stages of tumorigenesis. Thus, it is not surprising that TGFβ pathway has been shown to regulate metastatic dormancy in opposite directions according to cancer subtype, target organ and TGFβ ligand. TGFβ2 has been consistently reported as dormancy inducer. Inhibition of TGF-β receptors induces multi-organ outgrowth of disseminated HNSCC [110], proliferation of dormant prostate cancer cells in an in vitro model of dormancy in the bone marrow [111] and poor survival rate in prostate cancer patients [112]. Importantly, TGFβ2 expression was found to be high in single cells isolated from bone marrow of patients with dormant disease, while it dropped in patients with advanced disease. On the contrary, TGFβ1, together with POSTN and TNC, has been proposed as a mediator of reawakening of indolent cells disseminated in proximity of vascular sprouting tips in lung or bone marrow [83]. Other members of the TGFβ superfamily, bone morphogenetic proteins (BMPs), have been shown to induce dormancy as well. BMP4 from lung microenvironment induces dormancy of disseminated breast cancer cells and its inhibition by COCO drives metastatic outgrowth of the cells [113]. Similarly, BMP7 induced dormancy of prostate cancer cells in the bone marrow [114]. The cytostatic effect of TGFβ ligands has been connected to its well-known capacity to induce cyclin-dependent-kinases (CDKs) inhibitor genes such as p21 and p27 [110,111,112,114]. Alternatively, as in the case of epithelial cells, TGFβ stimulation can result in a quiescent phenotype as a consequence of a transdifferentiation process, such as epithelial-to-mesenchymal transition (EMT) and/or stemness. EMT is a physiological process, induced by growth factors (such as TGFβ or FGF), whereby epithelial cells activate specific transcription factors (such as Snail, Slug, Twist1, ZEBs) that cause disassembly of adherent junction complexes, reorganization of cytoskeleton to support invasive properties and ultimately gain of mesenchymal markers and loss of epithelial identity. This process is physiologically activated during embryogenesis and wound healing, but contributes to pathological conditions, such as cancer aggressiveness and fibrosis, when dysregulated [115,116]. While it is widely accepted that cancer cells at tumor edge are endowed with mesenchymal traits that are required for local infiltration and systemic dissemination [115,117], the role of EMT transcription factors into dormancy and reawakening is more nuanced. Different groups have reported that stable overexpression of EMT transcription factors Zeb1 and Twist1 leads to the outgrowth of dormant lung-disseminated breast cancer cells [118,119]. However, others experimentally demonstrated that Twist1 is sufficient to induce EMT and promote dissemination of cancer cells from primary tumor mass to secondary organs, but then Twist1 has to be switched off to allow metastatic outgrowth [120,121]. Possible solutions to these conflicting data are the reciprocal regulation of different EMT transcription factors [117] and their effect after continuous vs. dynamic expression; indeed, intermittent, but not stable, overexpression of Snail1 led to metastatic outgrowth of disseminated cancer cells [122]. Several groups reported that EMT might be accompanied by gain of stemness features by circulating cancer cells [117,123,124] and that dormancy of disseminated cells often correlates with cancer stem cells (CSC) phenotype [113,114,125,126]. However, complete switch between epithelial to mesenchymal phenotypes is rare [115] and data from patients support the idea that circulating and disseminated cells carry mixed epithelial and mesenchymal markers [127,128,129]. Recently, partial EMT was detected in early disseminated cells and correlated with survival in lungs and bone marrow [130]. It is possible that the choice between survival vs. death and proliferation vs. quiescence of disseminated cells is dictated by the frequency and/or ratio between different determinants of epithelial and mesenchymal phenotypes. In this scenario, local variations of TGFβ ligands availability in the metastatic niche is likely to play a significant role. Importantly, it has been reported that mechanical stress enhances release of TGFβ2 from a preformed pool in bronchial epithelial cells [131,132,133] and that stiffness of substrate regulates the choice between TGFβ-induced apoptosis or EMT [134].

As described above, matrix stiffness directly induces integrin clustering, focal adhesion assembly and maturation and subsequent activation of FAK- and RHO-dependent Ras/ERK pathway [12,30,96]. Of note, EGFR itself has been shown to be activated by compressive stress [135], and the capacity of EGF ligands to activate ERK is promoted by ECM stiffness [136]. While the ratio of two different MAPKs, namely ERK and p38, has been consistently reported as a determinant of the dormancy phenotype [95,137], how and if different mechanical stresses have different impact on this ratio is currently unknown. Finally, the mechano-responsive YAP/TAZ and MRTF pathways have been directly linked to the activation of metastatic spreading of disseminated cancer cells. Massague and colleagues showed that upon dissemination, metastatic cells spread on capillaries, dislodging resident pericytes, and activating YAP/TAZ and MRTF pathways [107]. L1CAM regulates this spreading, ILK activation and YAP nuclear localization. The role of YAP into metastatic outgrowth is further reinforced by a recent publication from Egeblad lab, where they showed that inflammation/smoke-induced NET deposition in the lung microenvironment triggers laminin cleavage, unmasking of integrin α_3_β_1_ stimulating epitope, and YAP activation [91].

## 9. Conclusions

The mechanical properties of the microenvironment likely play a fundamental role in many steps of the metastatic cascade. Despite significant technical advancements allowing for the evaluation of the relevance of several mechanical parameters to primary tumour growth, invasion and hematogenous dissemination in vivo and in vitro, their benefit for the understanding of metastatic dissemination and outgrowth have been limited so far. This is primarily due to the difficulty of modulating forces and the architecture at the single cell level, which is the scale at which disseminated metastatic cells integrate signals that ultimately lead to the proliferative choice (survival vs. death, quiescence vs. proliferation). Nevertheless, mechanosensing membrane proteins and intracellular transducers, as well as ECM proteins and stress-modulated ligands, have been involved in the metastatic phenotype, suggesting that forces surrounding disseminated cells are likely to have an important role in their fate. However challenging, the development of accurate in vitro models incorporating mechanical parameters and mimicking metastatic dormancy or growth might be a valuable strategy to overcome the aforementioned limitations, to enable the study of metastatic cells in more physiologically-relevant conditions, and thus lead to the discovery of new druggable targets for the eradication of disseminated metastatic cells.

## Figures and Tables

**Figure 1 cells-09-00250-f001:**
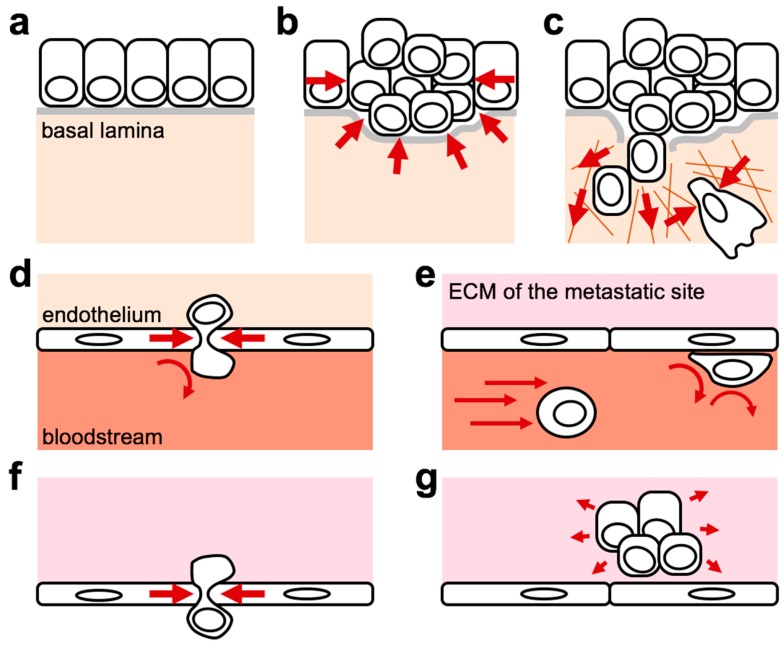
Cancer cells are exposed to different forces while they undergo metastatic dissemination. The scheme exemplifies the development of a solid tumor originating from an epithelium (**a**). Transformation and neoplastic growth may increase local crowding and intratumoral pressure (**b**), activating mechanical competition mechanisms, neoangiogenesis (not shown), and degradation of the basal lamina. Remodeling and stiffening of the extracellular matrix (ECM) in cooperation with cells of the stroma (not shown) provide higher resisting forces, which promotes cell tension, outward cell migration and sustain cancer cell survival, proliferation and tumour-initiating properties (**c**). Migration within a dense ECM may also cause compressive stresses, leading to DNA ruptures and activation of DNA-damage responses (**c**). Cells invading the local stroma may reach the vessels and intravasate by physical crossing the endothelial barrier (**d**). In the blood, disseminating cells lack of adhesion to the ECM and experience shear stresses that may influence their preferential homing to some target organs and the ability of cells to adhere to the endothelium (**e**). Once cells successfully pass the endothelium and extravasate (**f**), they reach a novel ECM microenvironment (**g**) with new mechanical properties (in the example, a softer ECM providing lower resisting forces and thus enabling cells to develop lower traction forces). Red arrows indicate external forces applied on cells by the microenvironment.

**Figure 2 cells-09-00250-f002:**
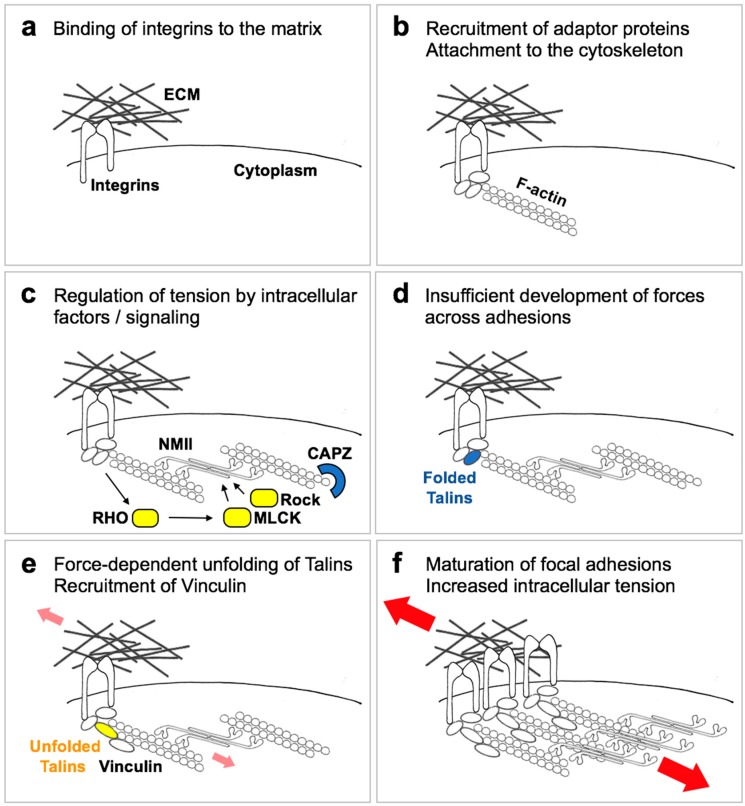
A simplified scheme of ECM mechanosensing. When cells adhere to an ECM via integrin receptors, cell-substrate adhesions are transiently formed (**a**). These adhesions are stabilized by recruitment of adaptor proteins to integrin cytoplasmic domains, which in turn connect to actin filaments (F-actin), leading to formation of focal points (**b**). Locally activated signaling proteins (**c**) regulate the development of myosin (NMII)-mediated tension, counteracted by negative regulators such as CAPZ. If cell-generated tension is not opposed by extracellular resistance (**d**), such as on a soft ECM, adhesions remain at this stage or disassemble. If instead a stiff ECM efficiently opposes resisting forces (small red arrows), tension across adhesion complexes induce the unfolding of adaptor proteins such as Talin1/2 and the recruitment of new players, including Vinculin (**e**). This enables the growth and reinforcement of cell-substrate adhesions that mature into focal adhesions (**f**), in turn sustaining the development of higher cell traction forces (large red arrows).

**Figure 3 cells-09-00250-f003:**
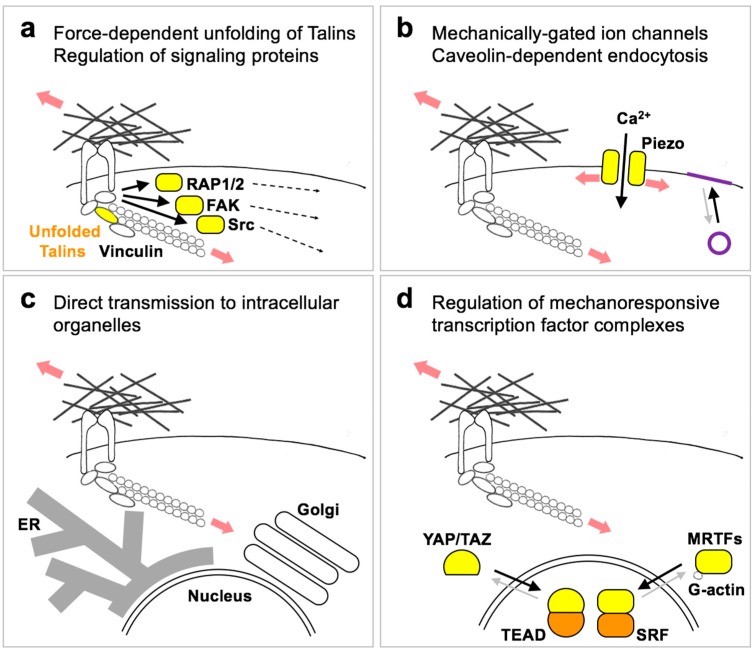
A simplified scheme of ECM mechanotransduction mechanisms. Resisting forces provided by ECM stiffness regulate multiple signaling mechanisms, including (**a**) direct force-dependent unfolding of proteins at adhesion structures (Talins), leading to recruitment of adaptor proteins (Vinculin) and to the regulation of signaling proteins such as GTPases (Rap1/2) and kinases (FAK and Src), which then relay the signal to the rest of the cell (dashed arrows). (**b**) Tension of the plasma membrane can induce the opening of mechanically-gated ion channels such as Piezo1/2, and may influence the rate of Caveolin-dependent endocytosis (indicated in purple) as a feedback mechanism to accommodate membrane stretching. (**c**) Extracellular tension can be directly transmitted to intracellular structures and organelles, including the nucleus, the endoplasmic reticulum (ER) and possibly the Golgi apparatus, regulating their functions (see text). (**d**) Extracellular forces also regulate gene transcription by activating transcription factors such as the YAP/TAZ-TEAD and MRTF-SRF complexes. regulation of YAP/TAZ entails multiple parallel mechanisms (see text), while MRTFs activity is directly regulated by the ratio between filamentous (F-actin) and monomeric/globular actin (G-actin).

**Table 1 cells-09-00250-t001:** Signaling pathways regulating disseminated metastatic cell fate.

Signaling Pathway	Biological Outcome
CollagenI (ECM stiffness)–Integrin–MLC–FAK/ERK/Src	Quiescence to proliferative switch at metastatic site
LOXL (Collagen crosslinking)	Metastatic efficiency
NETs–cleaved laminin–MLCK–YAP	Outgrowth of metastatic dormant cells
VWF/VCAM-1–Integrin	Resistance to chemotherapy of metastatic dormant cells
TNC – Notch/Wnt	Increased fitness of disseminated cells
TGFβ–fibrillar FN–Integrin–FAK/ERK	Metastatic cell survival
FN–ERK/p38 ratio	Quiescence to proliferative switch at primary site
Mucin-1–Integrin–PI3K/AKT	Increased fitness of disseminated cells
TSP-1	Quiescence of disseminated cells
POSTN–BMP1/LOX and Wnt	Metastatic cell outgrowth
Rif/mDia2 + ILK/β-Parvin/Cofilin-FLPs	Metastatic cell outgrowth
Fascin–Filopodia	Metastatic efficiency
L1CAM–Integrin–ILK–YAP/MRTFs	Metastatic cell survival and outgrowth
Collagen–DDR1–JAK/STAT3	Outgrowth of metastatic dormant cells
Syndecans–PAR polarity complexes-ERK	Metastatic cell quiescence
TGF-β superfamily	Ligand- and organ-dependent control of quiescence or proliferation of disseminated cells

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
