# Peer review of "Mechanical Forces as Determinants of Disseminated Metastatic Cell Fate"

_cells, 2020, doi:10.3390/cells9010250_

Round 1
Reviewer 1 Report
The authors tried to collect and summarise the knowledge available on mechanobiology of disseminated cells fate. As two third of the listed references are youger than ten years old, mostly the last decade is covered. The manuscript enumerates the most important achievments regarding mechanobiology of disseminated cacer cells, mostly focusing on signal transduction and molecular details. Although enormous work was done, few eye catching or summarising figures might enhance the perception and ease the delivery of the basic ideas about force driven processes in metastatic cell fate. Additionally, the Conclusion paragraph sounds a bit low in details compared the amount of information enlighted by the manuscript. Furthermore, in some cases the used abbreviations are not introduced, which hardens the understanding for non close - field readers. Attention should be payed to the style of the listed references as well, having them uniforme would ease the reading. Minor note, keywords are missing.
Author Response
We thank the reviewer for the positive comments about the manuscript and for the insights on how to improve it. Our initial idea was to provide a quick overview about an aspect of metastasis biology that we felt wasn’t yet properly addressed, i.e. the role of mechanical forces, with all the relevant references that support a role of mechanobiology at the metastatic site. As this, as well as the other, reviewer pointed out, this might have led to excessive compression of informations and references. Thus, we added four new items to help the reader following the logic flow of the main text:
Figure 1 about forces involved during the metastatic journey
Figure 2 about cell-ECM interaction during mechanosensing
Figure 3 on mechanisms of mechanotransduction
Table 1 with all the mechano-related proteins with a role in metastatic dormancy or outgrowth as well. For each of them we reported: mechanosensing molecule, mechanotransduction pathway and biological output.
We also added a complete list of the abbreviations used and corrected the style of references and main text. We hope that the reviewer will find the revised version improved and suitable for publication.
Reviewer 2 Report
Montagner and Dupont describe the current state of knowledge with respect to the topic. The paper is well written. The authors focus on the interaction of disseminated metastatic cells with their environment. They characterize/describe the different stages of mechanotransduction. In my opinion, the paper should be published with only very minor revisions (and some wishes from my side, if possible).
(1) Minor revisions:
Throughout the paper, likely due to a formatting problem, latin signs appear to be shown wrongly (e.g. line 453 for TGF-β). This has to be corrected. In the paper, there are a handful of minor typos (e.g. Line 399 “filpodia”, or Line 402 “we recently shown”).(2) Wishes: I think that the following will improve even more the manuscript:
Please, include a list of abbreviations (many are mentioned in the text). Please, add a figure summarizing the “force journey” described in the section “ECM mechanical forces in cancer”. A cartoon would greatly help non-expert readers to better understand the current state. Please, add a figure describing the different states of mechanotransduction, as mentioned in the chapter “Cells measure the physical properties of the ECM”. Again, a cartoon would help non-expert readers. In the chapters “mechanosensing receptors and metastatic fitness” and “mechanosensing pathways and dormancy at the metastic site” quite an excessive number of examples are given. A summarizing table (including the name of mechanosensor, function, target) will help the reader.
Author Response
We thank the reviewer for the very positive review of our manuscript. We have fully taken onboard the reviewer suggestions and now added four new items to help the reader following the logic flow of the main text:
Figure 1 about forces involved during the metastatic journey
Figure 2 about cell-ECM interaction during mechanosensing
Figure 3 on mechanisms of mechanotransduction
Table 1 with all the mechano-related proteins with a role in metastatic dormancy or outgrowth as well. For each of them we reported: mechanosensing molecule, mechanotransduction pathway and biological output.
We also added a complete list of the abbreviations used. We hope that the reviewer will find the revised version improved and suitable for publication.
Reviewer 3 Report
Mechanical forces as determinants of disseminated metastatic cell fate (review)
This review about metastasis and extracellular matrix is interesting and original and it could be interesting for this field.
Its greatest difficulties are because the information density, it has difficulties to follow the topic, since it lacks graphics and figures that help interpret and guide the work done. I think it will be an interesting contribution the same if the work is reorganized and used as graphic guides or figures that facilitate reading.
It would be interesting to divide the extracellular matrix events produced in the initiation of the process and in the progression of it. In each case indicate the mediators and possible events that occur in schemas.
Author Response
We thank the reviewer for the positive comments about our manuscript. As highlighted by the other reviewers as well, the original version of the review was likely hard to read due to density of references without schematics in support of the reader. For this reason, in the current version we added four new items to help the reader following the logic flow of the main text:
Figure 1 about forces involved during the metastatic journey
Figure 2 about cell-ECM interaction during mechanosensing
Figure 3 on mechanisms of mechanotransduction
Table 1 with all the mechano-related proteins with a role in metastatic dormancy or outgrowth as well. For each of them we reported: mechanosensing molecule, mechanotransduction pathway and biological output.
We also added a complete list of the abbreviations used. We hope that the reviewer will find the revised version improved and suitable for publication.